# Expression, Localization, and Activity of the Aryl Hydrocarbon Receptor in the Human Placenta

**DOI:** 10.3390/ijms19123762

**Published:** 2018-11-27

**Authors:** Anaïs Wakx, Margaux Nedder, Céline Tomkiewicz-Raulet, Jessica Dalmasso, Audrey Chissey, Sonja Boland, Françoise Vibert, Séverine A. Degrelle, Thierry Fournier, Xavier Coumoul, Sophie Gil, Ioana Ferecatu

**Affiliations:** 1National Institute for Health and Medical Research, UMR-S 1139, Faculté de Pharmacie de Paris, Université Paris Descartes, Sorbonne Paris Cité, F-75006 Paris, France; anais.wakx@free.fr (A.W.); margaux.nedder@outlook.fr (M.N.); jessica.dalmasso49@gmail.com (J.D.); audrey.chissey@parisdescartes.fr (A.C.); francoise.vibert@parisdescartes.fr (F.V.); severine.degrelle@inserm.fr (S.A.D.); thierry.fournier@parisdescartes.fr (T.F.); sophie.gil@parisdescartes.fr (S.G.); 2National Institute for Health and Medical Research, UMR-S 1124, Université Paris Descartes, Sorbonne Paris Cité, F-75006 Paris, France; celine.tomkiewicz-raulet@parisdescartes.fr (C.T.-R.); xavier.coumoul@parisdescartes.fr (X.C.); 3Centre National de la Recherche Scientifique UMR 8251, Unité de Biologie Fonctionnelle et Adaptative, Université Paris Diderot, Sorbonne Paris Cité, F-75013 Paris, France; boland@univ-paris-diderot.fr; 4Inovarion, 38 Avenue des Gobelins, Paris F-75013, France

**Keywords:** AhR, ARNT, benzo-a-pyrene, CYP1A1, CYP1B1, placenta, trophoblast, ontogeny

## Abstract

The human placenta is an organ between the blood of the mother and the fetus, which is essential for fetal development. It also plays a role as a selective barrier against environmental pollutants that may bypass epithelial barriers and reach the placenta, with implications for the outcome of pregnancy. The aryl hydrocarbon receptor (AhR) is one of the most important environmental-sensor transcription factors and mediates the metabolism of a wide variety of xenobiotics. Nevertheless, the identification of dietary and endogenous ligands of AhR suggest that it may also fulfil physiological functions with which pollutants may interfere. Placental AhR expression and activity is largely unknown. We established the cartography of AhR expression at transcript and protein levels, its cellular distribution, and its transcriptional activity toward the expression of its main target genes. We studied the profile of AhR expression and activity during different pregnancy periods, during trophoblasts differentiation in vitro, and in a trophoblast cell line. Using diverse methods, such as cell fractionation and immunofluorescence microscopy, we found a constitutive nuclear localization of AhR in every placental model, in the absence of any voluntarily-added exogenous activator. Our data suggest an intrinsic activation of AhR due to the presence of endogenous placental ligands.

## 1. Introduction

The placental barrier is a highly complex tissue, which regulates exchanges between the maternal and fetal circulations. This function is essential for the maintenance of pregnancy and for the development and growth of the embryo and then the fetus. The placenta is composed of villous epithelial, mesenchymal, and endothelial tissues, in which fetal vessels are embedded [1]. The epithelium of the villous tree is formed by the outer syncytiotrophoblastic layer, a multinucleated syncytium in direct contact with the maternal blood, and by the underlying mononuclear proliferative cytotrophoblasts, which constantly renew the syncytiotrophoblast layer. Because of its localization, the syncytium assures the synthesis and secretion of hormones, including steroids (progesterone, estrogens), human chorionic gonadotropin (hCG) placental growth hormone (pGH), human placental lactogen (hPL), and factors, like soluble fms-like tyrosine kinase-1 (sFlt1), involved in the regulation of blood vessel formation. The placenta also ensures the protection of the fetus against a wide variety of xenobiotics, including drugs and pollutants. However, the physiology of this placental barrier varies during ontogeny with major modifications occurring between early pregnancy and term. Although villous architecture and arborescence are organized at the end of the first month of pregnancy, they continue to branch out by sprouting in the area of basal decidua. At term, the villous surface area reaches as much as 12 to 14 m^2^ [2], is highly vascularized, and the tissue barrier is greatly thinned (1–3 µm as compared to 45 µm in the first trimester of pregnancy) [3]. Furthermore, major changes occur in the trophoblastic dioxygen (O_2_) environment within the first trimester, with an increase from 2–5% O_2_ (before the eighth and ninth weeks of amenorrhea (WA)) to 6–8% O_2_ (after 12–14 WA). All these characteristics need to be considered when investigating the impact of xenobiotics on placental functions since significant crosstalk may occur between several signaling pathways (e.g., adaptation to hypoxia and stress responses to pollutants). 

The aryl hydrocarbon receptor (AhR) is an environmental-sensor transcription factor that mediates the metabolism of a wide variety of xenobiotics, which are present in the environment and in food, including polycyclic aromatic hydrocarbons (PAH) and dioxins [4]. The high-affinity ligand-binding ability of the AhR and the downstream results of activation lead to the metabolic breakdown and elimination of many chemicals. However, the metabolic bioactivation of some chemicals contributes to adverse cellular toxicity [5]. The AhR is a cytosolic receptor located in the core of a multiprotein complex, which is composed of chaperones and kinases that protect and maintain the receptor in a conformation that can bind ligands. Upon ligand-binding, the AhR relocates to the nucleus, undergoes heterodimerization with the AhR nuclear translocator (ARNT), and binds to specific xenobiotic responsive elements (XRE) in the promoters of its target genes. This results in the initiation of gene transcription [6]. Among the target genes of the AhR are genes encoding xenobiotic metabolism enzymes (XME) of phase I, such as the microsomal cytochrome 450-dependent monooxygenases (CYP1A1, CYP1A2, and CYP1B1), and XME of phase II, like heme oxygenase 1 (HO1), NAD(P)H quinone dehydrogenase 1 (NQO1), superoxide dismutase 1 and 2 (SOD1/2), and Glutathione S-transferase (GST) [7]. Following ligand-dependent activation, the AhR signaling pathway is tightly regulated by several negative feedback loops. AhR is constitutively exported to the cytosol and degraded by the 26S proteasome pathway [8]. On the other hand, one of its target genes, the AhR repressor (AhRR), is a competitor for the AhR-ARNT dimerization (AhRR-ARNT) and for the subsequent binding to the XRE sequence [9].

Recent data suggest that AhR is involved not only in the detoxification process, but also in other main cellular functions, for instance, regulating the immune response, cell differentiation, the cell cycle, and oncogenesis [10,11,12]. These additional functions of the AhR have been brought to light by the discovery of numerous dietary ligands (e.g., indoles, carotenoids, flavonoids, resveratrol), microfloral and endogenous ligands (l-tryptophan derivatives) [13,14], and from pioneer studies employing *AhR*-knockout organisms that exhibit several developmental defects [15,16,17,18,19]. In addition, it has been shown that *AhR*-knockout mice exhibit alterations in fertility, decreased litter size, and increased neonatal death of offspring [20], which suggests that AhR has a critical role during pregnancy. Moreover, endogenous ligands, like kynurenine (a l-tryptophan-derivate metabolite produced by two enzymes: Indoleamine 2,3-dioxygenase 1 enzyme (IDO1) or tryptophan 2,3-dioxygenase (TDO)), which directly activate the AhR in immune cells from the decidua and peripheral blood, have been proposed to be essential for the maintenance of pregnancy, by allowing the immunological tolerance of the embryo by the mother [21,22,23,24,25,26]. The activation of the AhR-dependent CYP1A1 pathway by environmental pollutants has been associated with profound immunosuppression [27] and complications of pregnancy (e.g., spontaneous-abortion, intrauterine growth restriction, premature birth) [28]. 

Previously published results indicate that the AhR is highly expressed in human placenta, at the periphery of the chorionic villi in syncytiotrophoblasts (ST) and also in the endothelial cells of villous large-blood vessels and umbilical cord veins/arteries [29,30]. However, the expression of the AhR in the human placenta has been demonstrated mostly at the mRNA level and the localization of the protein only by immunohistochemistry. At present, a complete temporal (from early pregnancy to term) and spatial (within the tissue) description of the expression at the protein level is lacking.

In this study, we have established the cartography for protein expression, localization, and intrinsic activity of the AhR, and we have compared these data among different placental models: Explants of chorionic villi at different periods of pregnancy, cultures of primary trophoblasts that differentiate into syncytiotrophoblasts in vitro, and in the BeWo cell line in respect to its confluence state. We also assessed the AhR intrinsic activity measuring the expression of several target genes. Then, we investigated AhR localization and activity after incubation of BeWo cells with Benzo(a)Pyrene (BaP), a strong AhR ligand. This foundational study is essential for a better understanding of the AhR distribution and role—detoxification and/or physiological functions—in the placental barrier to better assess the impact of environmental pollutants at the maternal-fetal interface.

## 2. Results

### 2.1. AhR Expression in Human Placenta During Ontogeny

We determined the pattern of AhR expression in the placenta throughout pregnancy. To this end, placental villi from 8–9 WA, 12–14 WA, and 37–39 WA (term) were harvested and used for both RT-qPCR and Western blot analysis. To verify that there were no ex vivo modifications, such as tissue degradation, during dissection or RNA extraction, we first assessed the level of typical placental biomarkers, such as *β-hCG, hPL, PLGF,* and *GH2*, and enzymes, such as *CYP11A1* (P450ssc) and *CYP19* (Aromatase), responsible for the production of placental-derived progesterone and estrogens, respectively [31]. All data were normalized to the average of three housekeeping transcripts (*HPRT*, *RPL13*, and *18S*) and then normalized to the level of *KRT7 (Cytokeratin-7, CK7)* mRNA, an intermediate filament protein expressed in trophoblasts, but not in other placental cell types. The normalization to CK7 is needed to take into account the trophoblast mass variation within the placental villi during placenta development and growth, and the variability of trophoblasts’ mass between the samples. As expected, we found (Figure 1A) that *β-hCG* exhibits a higher expression (the average Ct is 20) in the early first trimester of pregnancy as compared to 12–14 WA and term conditions (mean Cts of 22 and 28, respectively) [31]. We also found that the amounts of *hPL*, *PLGF*, and *GH2* as well as *CYP11A1* and *CYP19* mRNAs were increased at term. Then, we designed primers specific for the human AhR, the AhR repressor (AhRR), and their partner, ARNT, and validated them by PCR with total RNA extracted from human placental primary trophoblasts (Appendix A). The transcripts of *AhR*, *AhRR*, and *ARNT* are expressed at a low level (mean Cts of 29, 32, and 32, respectively, Figure 1B) and remain unchanged within the first trimester of pregnancy (8–9 WA and 12–14 WA). The expression of *AhR, AhRR,* and *ARNT* are all increased in term placental villi when normalized to KRT7, as much as five-fold for both *AhR* and *AhRR* (respectively, average of 1.09 compared to 5.09; 1.06 compared to. 4.41) and more than two-fold for *ARNT* (average of 1.05 compared to. 2.34). In order to determine whether the increased amount of *AhR* mRNA is followed by an increase in the level of protein, we performed Western blots with protein extracts of placental villi from different stages of pregnancy. The amounts of AhR and ARNT protein were significantly increased at 37–39 WA (Figure 1C and right bar graph for total AhR quantification) as compared to the first trimester (8–9 WA and 12–14WA). The specificity of the two bands of AhR (at around 95 and 130 kDa) was checked using a specific blocking peptide for the antibody (Figure 1C). In eight different term placental villi extracts, we observed an interindividual variability in the level of AhR of 0.5 to 3 arbitrary units (AU) (Appendix A). Finally, we found no major variation in the amount of AhR and ARNT protein in different regions of term placenta after sampling villi from central (close to the umbilical cord), intermediate, and peripheral zones (Appendix A).

Since the AhR is expressed differently during ontogeny, we next investigated whether its activity is also enhanced when its expression level is increased at 37–39 WA. We evaluated by RT-qPCR, the mRNA expression levels of several AhR target genes. Compared to the amount of mRNA in the first trimester, the expression of CYP1B1 is not modified during ontogeny (Figure 1D). In contrast, the amounts of *CYP1A1* and *CYP1A2* mRNA were four and six-fold higher (respectively, average of 1.33 compared to. 4.21; 1.58 compared to. 6.29). These results suggest that, in the absence of any voluntarily-added exogenous ligands, AhR is activated in term placenta. To further pursue this observation, we also evaluated the expression of several other XME AhR-transcriptional targets, which are involved in phase II of detoxification. The expression of these genes was compared in villi from the first trimester (8–9 WA and 12–14 WA) to villi from 37–39 WA placenta. The mRNA of all AhR anti-oxidative targets (*HO1, SOD1, SOD2, NQO1, GSTA2,* and *Catalase* mRNAs) were significantly increased at term as compared to the first trimester of pregnancy (Table 1). Taken together, these results demonstrate that the expression and the activity of the AhR is increased at term.

### 2.2. AhR Localization in Human Placental Chorionic Villi During Ontogeny

Next, we examined the localization of AhR in chorionic villi of placentae from different periods of pregnancy by confocal microscopy. First, the immunostaining conditions were optimized by testing several anti-AhR antibodies (WH0000196-M2, Ab2770, A9359, HPA029722, and HPA029723) along with a non-specific IgG negative control, under different fixation/permeabilization conditions in both villous tissues and trophoblast cultures (primary and BeWo) (Appendix A). Optimal staining of AhR in cells and tissue was obtained with the HPA029723 and WH0000196-M2 antibodies. Chorionic villi from 8–9 WA, 12–14 WA, and 37–39 WA were stained with the HPA029723 AhR antibody to determine AhR localization in the different types of cells of the tissue. To identify the trophoblast layer(s), an anti-cytokeratin 7 antibody was tested in parallel to the staining of nuclei with DAPI. AhR was detected in both trophoblast cells and mesenchymal cells of chorionic villi from all periods of pregnancy (Figure 2A). Staining was absent when the AhR-antibody was pre-incubated with its blocking peptide (Figure 2A) or with the IgG controls (Appendix A), which were used here to determine the specificity of the antibody. The AhR was present mainly in the nucleus of trophoblast and mesenchymal cells, as observed by the superposition with DAPI-staining. Then, we determined in which type of placental cells, other than trophoblasts, the AhR might be expressed. Immunoblotting showed the presence of the AhR in protein extracts from mesenchymal cells (MC), umbilical arteries endothelial cells (A), heterogeneous cells (choriotrophoblasts, fibroblasts, and mesenchymal cells) (HC), endothelial progenitor cells (EPC), and umbilical vein endothelial cells (V) (Appendix A). CD31 was used as marker of endothelial extracts and β-Actin as a loading control.

In most of the non-pathologic tissues that have been examined previously, AhR was found to be localized to the cytosol and redistributed to the nucleus in the presence of ligands. Because our results showed a predominant nuclear localization of AhR in placental cells, without any voluntarily-added ligand of AhR, and because the AhR is present in several types of placental cells, we then examined the sub-cellular distribution of the AhR in freshly harvested chorionic villi following cell fractionation of the whole tissue. Reactivity to Lamin A/C nuclear protein and cytosolic α-Tubulin antibodies was used to assess the purity of the nuclear-enriched and cytosolic fractions (Figure 2B). Clearly, the two bands corresponding to the AhR are detected mostly in the nuclear-enriched fractions (89%) with only a very slight cytosolic presence (11%). These observations led us to consider that the AhR is activated and localized in the nucleus of chorionic villous cells in the absence of added exogenous ligand and to suspect the presence of some endogenous ligands.

Recently, kynurenine (kyn), an l-tryptophan (l-trp) derivative, was identified as a ligand and an activator of the AhR in several models, including placenta. Therefore, we determined the protein profile of two enzymes involved in the first step of L-trp metabolism, namely indoleamine 2,3-dioxygenase-1 (IDO1) and tryptophan 2,3-dioxygenase (TDO), in chorionic villi extracts from different periods of pregnancy. The IDO1 protein level is significantly increased in term villi, as is the AhR protein level (Figure 3). IDO1 slightly varies in different placentae at term (Appendix A), but less so than within different areas of the placenta (Appendix A). In contrast, the TDO enzyme was undetectable by Western blot (Figure 3), in agreement with other publications, which suggest its lack of expression in placental tissues.

### 2.3. AhR Expression, Localization, and Activity in Human Purified Trophoblasts

Human purified trophoblasts, which can be further differentiated into syncytiotrophoblasts (ST), permit investigation of the profile of AhR expression over the course of differentiation. Freshly purified human term cytotrophoblasts were immediately used (VCT_0_) or cultured to ultimately form ST after 72 h of in vitro culture, with an intermediate state VCT at 24 h. The level of *AhR* mRNA was unchanged during trophoblast differentiation (VCT_0_ vs ST) (Figure 4A). However, the total amount of AhR protein (the two bands of AhR) was slightly decreased in ST as compared to VCT_0_ and VCT (undifferentiated VCT after 24 h of culture) (Figure 4B blot and lower left graph). However, there was an increase of the larger isoform of AhR (p130-AhR) in ST (Figure 4B blot and lower right graph), which may be due to post-translational modification in ST, slowing down protein migration in SDS-PAGE. After optimization of the fixation/permeabilization conditions in purified trophoblasts (Appendix A), we stained AhR in both VCT and ST cells together and co-stained with antibody to E-cadherin (a membrane marker) and nuclear DAPI. We observed the predominant nuclear localization of AhR (Figure 4C), similar to the chorionic villi (Figure 2A). IgG was used again as a negative control (Figure 4C lower images). This nuclear immunolocalization of AhR is not modified by cell fusion when forming the syncytium. This is visualized by the E-cadherin staining of the ST membranes containing several nuclei as highlighted with the dotted white line in the overlay (Figure 4C). 

Finally, we found that the mRNAs of the AhR main’s targets, *CYP1A1* and *CYP1B1*, were both increased when trophoblasts were differentiated to form the ST (about 100-fold change), although CYP1A2 mRNA decreased (Figure 4D). Altogether, our results indicate activation of AhR during differentiation.

### 2.4. AhR Expression, Localization, and Activity in the BeWo Trophoblast Model

The BeWo choriocarcinoma cell line is a model of trophoblasts that is often used in toxicological studies. We next determined whether this cell line could be used as a model to study the mechanism of the AhR pathway. To this end, and since AhR expression and activity have been demonstrated previously to vary according to the confluence level [32], we evaluated the levels of *AhR*, *AhRR*, and *ARNT* mRNAs as a function of the level of confluence of the BeWo cells. We found that the amount of the *AhR* mRNA increased (Ct 27 at 50% confluence) whereas expression decreased when the cells become subconfluent (Ct 28 at 90% confluence) (Figure 5A) and the level of AhR protein, when evaluated by Western blot, also is maximal at 50% confluence and follow the same kinetic (Figure 5B blot and right panel). In contrast, there was no significant variation in the amount of *AhRR* mRNA nor in the levels of ARNT mRNA and protein as a function of the state of confluence of BeWo cells (Figure 5A middle and right plots and Figure 5B). Following the optimization of experimental conditions using different fixation/permeabilization conditions and anti-AhR antibodies (Appendix A), we assessed the localization of AhR in BeWo cells in the absence of any treatment. IgG was used in parallel as a negative control. Similarly, to the previous experiments, the AhR (HPA029723) exhibited a solely nuclear localization in the BeWo cells (50% confluence) and AhR immunofluorescence was absent when a blocking peptide was employed (Figure 5C). BeWo cells were then fractionated to separate nuclei from cytosols and Western blotting was used to evaluate the presence of the AhR. α-Tubulin and Lamin A/C antibodies were used to assess the purity of the fractions. The two bands of AhR are detected mainly in the nuclear fractions (Figure 5D), which suggests that this localization is similar in BeWo cells and purified trophoblasts. Next, we found that the expression of *CYP1A1*, *CYP1B1*, and *CYP1A2* mRNA during BeWo proliferation diminished when cells become confluent (Figure 5E) as did the expression of AhR (Figure 5A). Moreover, the amount of *HO1*, *SOD1*/2, *NQO1*, *Catalase*, *GSTA2*, *GSS* (glutathione synthetase), and *NRF2* mRNAs, which are AhR-dependent genes and encode phase II XME, were also decreased at confluences greater than 50% (70% or 90%) (Table 2). Altogether, these results suggest that both AhR expression and activity depends on BeWo cells’ density.

### 2.5. Effect of Benzo-(a)-Pyrene on AhR Activity in the BeWo Model

BeWo cells were incubated with 1 µM benzo-(a)-pyrene (B(a)P), a concentration close to the B(a)P dose found in the systemic circulation or detected in the placenta of smoking mothers [33,34]. Cells were incubated from 1h to 24 h, with or without a pre-incubation with a classical AhR-inhibitor, CH223191, a potent and specific AhR antagonist. This was used to observe if the AhR goes to the cytoplasm in the presence of an antagonist of AhR. Epifluorescence microscopy (Figure 6A) showed that 3 h incubation with B(a)P did not modify the AhR-nuclear distribution, even in the presence of the AhR inhibitor, CH223191 (3 µM), as shown in the AhR/DAPI overlay images (Figure 6A, right panel). Further, the two bands of AhR are found mostly in the nuclear fractions (lanes 1, 5, 9, and 13) following cell fractionation of BeWo cells incubated or not with B(a)P for 1h or 24 h. Again, the pre-incubation of the cells with the AhR-inhibitor, CH223191, did not change the distribution of AhR, which was still found in the nuclear fractions (lanes 3, 7, 11, and 15).

We then analyzed the AhR signaling pathway of BeWo cells by RT-qPCR after incubation of the cells with 1 µM of B(a)P for 24 h. First, we found that the amounts of *AhR*, *AhRR*, and *ARNT* mRNA did not change after the incubation with B(a)P as compared to the solvent control (Co-DMSO) (Figure 7A). Then, the main target genes of the AhR were evaluated after incubation with B(a)P in the presence or absence of the AhR inhibitor. Despite the location of the AhR in the nucleus, Figure 7B shows that 1 µM of B(a)P was sufficient to induce a high level of expression of *CYP1A1* (an 80-fold change). This induction is almost, but not totally, eliminated by the pre-incubation with the AhR-antagonist (decrease from 80- to a 1.5-fold change). Similarly, *CYP1B1* and *CYP1A2* were induced by B(a)P (3-fold and 12-fold change, respectively). This induction was totally eliminated by the AhR-antagonist in the case of *CYP1A2*.

Finally, we also evaluated the amount of AhR protein in chorionic villi explants from term placentae after incubation with B(a)P for 24 and 48 h. There was no change in the amount of AhR protein even after 48 h of incubation with B(a)P (Figure 7C). There was a significant decrease in the level of IDO1 protein after incubation with B(a)P for 48 h, which has not been previously described (Figure 7C). Altogether, B(a)P exposure leads to the induction of a detoxification system and to a decrease in IDO1, and consequently impacts AhR physiologic activity in the placenta.

## 3. Discussion

It is generally accepted that the AhR is activated by a variety of environmental pollutants, such as dioxins and PAHs. In addition, roles for the AhR, other than mediating the toxic effects of xenobiotics, are emerging. These new roles involve metabolism, tumor-promotion, and the regulation of immunity [35]. The identification of diverse dietary and endogenous ligands, which are activators of the AhR in pathways that govern physiological functions in the liver and the immune system, opens new and interesting perspectives for the possible implication of AhR in these pathways in the placenta. For instance, the activation of AhR by endogenous ligands seems to be involved in the proper embryologic development of mice, since AhR-knockout mice displays severe phenotypic abnormalities, including developmental defects [20]. AhR activation recently has been correlated with innate and adaptive immune system regulation of the immune cells present within the decidua [21,36]. Further, several recent publications have underlined the potential involvement of abnormal AhR activation in unexplained miscarriage [37,38].

In the current study, we investigated the expression of AhR in placenta of first trimester and term from healthy pregnancies. Insight into the second trimester is lacking here due to the inaccessibility of these healthy placentae (French legislation limits voluntary abortion to 14 WA). We present evidence that the expression of both AhR mRNA and protein (using a specific anti-AhR antibody that does not recognize AhRR or ARNT) varies markedly during the ontogeny of the placenta with an increase in expression in term placentae (up to five-fold changes) as compared to first trimester placenta. Moreover, the increase in AhR expression at term is accompanied by that of ARNT, its nuclear partner required for promoter activation. Increased AhR expression correlates indeed with an increase in transcriptional activity as manifested by the upregulation of the AhR target gene batteries of both phase I (*CYP1A1* and *CYP1A2*) and phase II (*HO1*, *SOD1*/2, *NQO1*, *GSTA2*, and *Catalase*) metabolism of xenobiotics. We also demonstrated simultaneous increase in *AhRR* mRNA expression, *AhRR* also being a target gene of the AhR signaling pathway. All the results we obtained in this study are summarized in Figure 8. One interpretation of the increased expression of AhR at term relates to the extent of the cytotrophoblast layer, underneath the ST, as compared to the thickness of the placental barrier during ontogeny and defense mechanisms [39]. It is possible that a thick VCT epithelium could act, itself, as a physical barrier during the first trimester (and even more so when trophoblasts plugs within the spiral arteries block the blood supply of the intervillous chamber). In term placentae, however, the VCT layers of chorionic villi become dramatically thinner (20x) and a different defensive mechanism will be instituted, which could include a more efficient detoxification process resulting from an increased expression of AhR.

However, the increased expression of key members of the AhR signaling pathway in the placenta of humans during pregnancy is in slight discrepancy with that of AhR expression during ontogeny in rat models. In rats, it has been reported previously that both *AhR* and *ARNT* mRNAs peak between 15–18 gestational days (gd), and then diminish up to term (21 gd) [30]. These increases were correlated with the induction of *CYP1A1* mRNA at 18 gd in rat placenta, however, no other AhR-targets were evaluated in the study [30]. Our results, thus, underscore the importance of taking into account species-specificity when studying AhR expression and activity, especially during the ontogeny, since notable structural differences within the trophoblast barrier are known to exist between humans’ and rodents’ placentae (the latter has three trophoblast layers).

Another important point is that, in our study, the changes in the transcriptional activity of the AhR in the chorionic villi tissue during ontogeny are rather modest (five- to seven- fold changes for *CYP1A1* and *CYP1A2*) when compared to the typical acute CYP450 induction by pollutants (around 100-fold change) [40,41,42]. This is in accordance with previously reported data showing that both *CYP1A2* and *CYP1B1* transcripts were expressed at low levels in human placenta at term [43,44]. The level of AhR activity after a 24 h incubation with B(a)P was found to be increased as much as 750-fold for *CYP1A1* expression in the BeWo trophoblast cell line [45]. However, in our work, the levels of AhR activity in non-pathological term placentae of non-smoking women were close to that induced by resveratrol (a natural ligand of AhR present in food [46]) in a study using human term placental trophoblasts (around five-fold changes for both *CYP1A1* and *CYP1A2* mRNAs) [30]. Again, in the rat placenta, the intrinsic activity of AhR toward CYP1A1 induction was also moderate at 18 gd [30]. These results suggest that AhR is constitutively and moderately activated by an unknown mechanism in term placenta as compared to first trimester placenta.

Although, constitutive activation of the AhR could occur when its level rises, as previously described [47]. The presence of ligands from the diverse sources to which pregnant women unavoidably are exposed could also explain the increased expression of AhR target genes. For example, pollutants may bypass other biological barriers (pulmonary, intestina), or compounds in the diet or nutritional supplements may reach systemic circulation and, thus, the placenta. However, AhR activation could also arise from changes in the levels of endogenously produced ligands arising from placental metabolic activity. We found that protein expression of IDO1 enzyme follows a similar pattern to that of AhR from the first to third trimesters (Figure 8). IDO1 is a tryptophan catabolizing cytosolic enzyme, which plays a primary role in maternal-fetal tolerance by stimulating the differentiation of T lymphocytes and, thus, allowing immunotolerance [26]. In the placenta, IDO1 catabolizes the l-tryptophan to produce kynurenine, an endogenous AhR ligand [24], which may trigger the generation of Treg cells within the decidua through the binding and direct activation of AhR. It has indeed been observed in mice cells that kynurenine induce the generation of Treg through AhR activation [48]. Recently, Walker et al. [49] demonstrated an increased kynurenine production (around 200 ng/mg of tissue) at term as compared to the first trimester (30 ng/mg of tissue) in placental explant cultures. These results are in agreement with our results concerning the levels of IDO1 protein during pregnancy. It is important to note that we have attempted to exclude technical issues that could be responsible for an artificial induction of IDO1 in term placentae during tissue dissection (from direct contact with blood that contains kynurenine). Tissue dissections are performed rapidly (under 30 min) and, after several washes in HBSS media, are frozen in liquid nitrogen. The same protocol was applied for first trimester placentae, where IDO1 was not induced (Figure 3).

We observed different levels of AhR protein expression in term placentae of different individuals even after excluding the possibility of area-dependent internal variation of AhR levels within the placental tissues (Appendix A). Inter-individual variation in AhR mRNA levels has been observed in human placentae [30]. This could be the result of exposure of the individual to environmental toxicants (from ambient pollution and/or food), to food components (cruciferous plants contain AhR ligands) [46,50], or the result of differing IDO1-expression/activity between individuals. In a recent transcriptomic study, increased expression of IDO1 mRNA (around three-fold changes) was observed in male versus female cytotrophoblasts [51], which suggests that sex-dependency also should be considered when analyzing gene expression of human placental villi. However, no sex-variation in AhR expression has been found so far. Finally, inter-individual variation could also be explained by exposure to non-AhR ligands, which, nevertheless, modify AhR levels, as previously demonstrated in the mouse brain after in utero exposure to bisphenol A [52].

Immunocytochemical studies of mouse hepatomas revealed that the AhR is predominantly a cytosolic receptor under normal conditions and that the AhR relocates to the nucleus after exposure to xenobiotics, such as TCDD [53,54]. Cytosolic AhR is bound to a multi-protein chaperone complex in a conformation that allows ligand binding [6,13]. However, AhR has been found to shuttle between the nucleus and the cytosol in the absence of exogenous ligands. Although the binding of ligands increases the rate of nuclear import of AhR, it does not eliminate nuclear export of AhR [54]. In the present study, we found that AhR was localized to the nucleus in all of the models studied, which included purified primary trophoblasts (in the undifferentiated cytotrophoblasts and in differentiated syncytiotrophoblasts), the BeWo cell line, and placental chorionic villous cells (Figure 8). Our evidence of AhR nuclear localization is supported by the use of a series of controls, such as the testing of several commercial antibodies as well as by the optimization of experimental protocols. The localization of AhR in the nucleus is also observed in freshly harvested tissues that have been fixed and included in agarose. Thus, the influence of culture conditions, which can interfere with the localization of AhR, are excluded in this case. Since false positive staining may occur upon immunolabeling of AhR, we performed additional experiments employing cell fractionation in order to confirm the cellular localization of AhR. We also used phosphatase inhibitors in our fractionation study, as AhR-dephosphorylation could modify its localization during the experimental protocol. Evidence for the nuclear localization of AhR using immunohistochemistry (IHC), solely in the syncytiotrophoblasts of chorionic villi of both first and third trimester placenta, also was mentioned by Stejskalova et al. [30]. In order to better understand the activity of AhR in the nucleus, it would be of interest to further investigate the formation of complexes with its known partners, such as ARNT or other nuclear proteins, by performing chromatin immunoprecipitation in placental tissues and trophoblasts, in the absence of any voluntarily-added exogenous ligand.

The nuclear localization of AhR in our studies could be due to the presence of some endogenous ligands produced by placental tissue, such as kynurenine, which are present, albeit in low amounts, from the first trimester of pregnancy as has been described by Murthi et al. [49]. This is in agreement with our results of an AhR nuclear localization in other placental cell types, such as fibroblasts and endothelial cells (Appendix A). In a paper by Jiang et al. [29] using IHC, the authors concluded that AhR could also be expressed in some large vessel endothelial cells and in umbilical cord vessels. The AhR in endothelial cells could participate in the regulation of normal placental vascular functions, such as vascular remodeling, vasoconstriction, and vasodilatation, as observed in mice [55]. Interestingly, a permanent shear stress by the flow of blood through placental villi can also explain the activation of AhR as demonstrated by Conway et al. [56] in human umbilical endothelial cells (HUVEC and HAEC cell lines). These results correlate with the results that we obtained with our placental models. We have concluded that AhR is located in the nucleus of both undifferentiated cytotrophoblast and syncytiotrophoblast (Figure 4C). The nuclear localization of AhR also was observed in BeWo cells and the use of CH223191 AhR antagonist did not led to a cytosolic redistribution of AhR (Figure 6A). Taken together, the nuclear location and the constitutive activity of AhR at term suggest that AhR contributes to some physiological function in the placenta.

When we analyzed the amount of AhR protein by Western blot, we observed, mainly, the presence of two bands (p95 and p130). These bands were specific to AhR since the use of a specific blocking peptide (in excess) for the C-terminal epitope of the AhR antibody led to the absence of all bands of AhR in both chorionic villi (Figure 1C) and BeWo (Figure 5B) protein extracts. The two-band profile of AhR was also observed in extracts of nuclear-enriched fractions, after cell fractionation, and was not modified when the AhR inhibitor, CH223191, was applied to the cells (Figure 6B). The antibody (epitope between amino acids 721–821 in the C-terminus) used in our Western blots does not recognize either the AhRR or the ARNT proteins, both of which have some degree of protein homology with AhR within the *N*-terminus. The appearance of a doublet of AhR (104 and 106 kDa) on Western blots from several cell lines has been mentioned before in the literature. The bands were not modified by the use of a β-naphtoflavone or TCDD and were probably not due to proteolysis during homogenization protocols. They were attributed, potentially, to two forms of AhR in human cells [57]. However, we observed that the p130 AhR band was increased when VCT differentiated into ST (Figure 4B and Figure 8). This was accompanied by an increased transcriptional activity of AhR in ST as was also observed with both CYP1A1 and CYP1B1 (Figure 4D). This could be due to some post-translational modifications of AhR, which occur when cells differentiate to ST, such as phosphorylation, ubiquitination, and SUMOylation, all of which are known to affect the status and activity of AhR. Post-translational modifications of AhR have already been described in human and hamster cells: Phosphorylation/dephosphorylation of the AhR at Ser12 and/or Ser36 residues, in proximity to the nuclear localization signal (NLS) or nuclear export signals (NES), were associated with a cytosolic versus nuclear AhR distribution, respectively, and with AhR activation [32,47,58]. Whereas ubiquitination of AhR promotes its degradation by the proteasome pathway [59], the SUMOylation of K63 and K510 has been demonstrated to enhance AhR stability [60]. In our work, the difference in size between the two AhR bands (p95 vs p130), their modulation during differentiation of VCT to ST (increased p130 band) (Figure 4B), and the difference in AhR-activity (*CYP1A1* and *CYP1B1* induction) (Figure 4D) suggest that post-translational modifications may finely modulate AhR transactivation when it is localized in the nuclei of trophoblast cells. We plan to address these aspects in a future work dealing with nuclear localization, expression, and activity.

Finally, in the BeWo trophoblast model, which is often used when addressing the effects of pollutants on the placental barrier, we observed that the level of AhR and its transcriptional activity were a function of the density of the cells (Figure 5A,B,E). Cell density has previously been described to influence and regulate the intracellular localization of AhR as well as its activity after exposure to pollutants [32]. We demonstrated here that exposure of BeWo cells to a strong ligand, such as B(a)P, did not modify AhR nuclear localization, either after short term exposure or after longer exposure periods (3 h and 24 h) (Figure 6). The AhR antagonist, CH223191, neither changed this nuclear localization. The intrinsic activity of the AhR toward *CYP1A1* was inhibited significantly by CH223191 (Figure 7B) alone, which reinforce the hypothesis of an endogenous activator of AhR in BeWo cells. Nevertheless, AhR activity was strongly increased by incubation of cells with 1 µM of B(a)P as compared to the induction of CYP450 gene transcription, which were partially (*CYP1A1* and *CYP1B1*) or totally (*CYP1A2*) inhibited by incubation with an AhR inhibitor (Figure 7B). In a next work, it will be interesting to check other AhR-inhibitors, such as alpha-naphtoflavone, 1,3-dichloro-5-[(1E)-2-(4-chlorophenyl)ethenyl]-benzene, or 6,2′,4′-Trimethoxyflavone, because they have various selectivities of antagonisms.

Seven types of PAH pollutants have been detected in a cohort of 200 pregnant women who had normal outcomes. Amounts as high as 6.15 ng/g dry weight placenta of B(a)P were found [33]. PAHs have been shown to be retained in the placenta and to be implicated in complications of pregnancies [61]. We show here that incubation with B(a)P has important consequences for IDO1 protein stability (Figure 7 and Figure 8). This could be due to the fact that IDO1, as a heme enzyme, is subject to redox control by the oxygen environment [62] and AhR-dependent metabolization of B(a)P leads to ROS generation. Thus, we hypothesize that pollutants, such as B(a)P, may interfere with some endogenous activity of AhR in the placenta. Pollutants switch the activity of AhR toward their own metabolism instead of those of endogenous compounds, such as retinol, for example. Similar reasoning has been put forward to explain how elevated levels of aromatic hydrocarbons may lead to impaired placental functioning by reducing the endocrine (gonadotropin, estrogen receptor, and hPL) and metabolic (glutathione transferase, phosphatase, and LDH) activities of the placenta [63].

## 4. Materials & Methods

### 4.1. Placentae Collection and Chorionic Villi Isolation

Placentae were collected from non-smoking, healthy women with pregnancies either delivered from elective and legally terminated pregnancies (during the first trimester between 8–9 and 12–14 weeks amenorrhea (WA)) or by Caesarean sections (between 37 and 39 weeks amenorrhea, hereafter called “term”). Placentae were obtained from the Port-Royal Maternity, the Mutualist Institute Montsouris, the Private Hospital of Antony, the Beclere Hospital, and the Beaujon Hospital after obtaining written consents from informed patients and approval from our local ethics committee (CPP: 2015-May-13909). The study was performed according to the principles of the Declaration of Helsinki. Placentas were obtained with the patients’ written informed consent. After collection, placental tissues were washed in Ca^2+^- and Mg^2+^-free Hanks’ Balanced Salt solution (HBSS, Gibco #14175, ThermoFisher, Illkirch, France). Then, chorionic villi were gently scraped free from vessels and connective tissue, and dissected into about 25 mg fragments and either frozen in liquid nitrogen, and then stored at −80°C for later total RNA and protein extraction; or fixed in 4% paraformaldehyde at 4 °C for 4h and then kept in PBS buffer for further immunolabelling assays. The time for placental dissection was kept under 30 min, and tissues were snap frozen with liquid nitrogen to limit tissue degradation [31].

### 4.2. Explants

Random harvested villi from term placentae, dissected as above, with a similar weight, were hung on needles and completely immerged in OptiMem media (Phenol Red-free, Gibco) as hanging villi, for 24 and 48 h, in 24-wells plates (Corning, Schiphol-Rijk, Netherlands) developed by SA Degrelle as previously described [64].

### 4.3. Cell Culture

The mononucleated villous cytotrophoblasts (VCT) were isolated, based on the methods of [65] and in a modified version of [66]. After dissection, the chorionic villi were washed in Ca^2+^, Mg^2+^-free HBSS, and then digested in trypsin-digestion medium (HBSS 5 mL/g containing 0.1% trypsin (Sigma-Aldrich #27250-018, Saint Quentin Fallavier, France), 0.1 M MgSO_4_ (Merck #5886-0500), 0.1 M CaCl_2_ (Merck #1-02820-1000, Fontenay sous Bois, France), 4% milk, and 50 Kunitz/ml DNAse type IV (Sigma-Aldrich #D5025) for 30 min at 37 °C without agitation. This was repeated eight times with 10 min incubation with the same trypsin solution. The three first trypsin digestions, containing a mix of extravillous cytotrophoblasts and VCT, were discarded after light microscopic analysis and the three last ones, containing a majority of VCT, were kept and pooled. The chorionic villi were washed finally with warm HBSS (37 °C). Each time, the supernatant containing VCT was collected after tissue sedimentation, filtered (40-µm pores), and incubated with 10% FCS (vol/vol) to stop trypsin activity. After purification by Percoll gradient, VCT were resuspended and were cultured in DMEM supplemented with 10% FCS, 2 mM glutamine, 100 IU/mL penicillin, and 100 µg/mL streptomycin (Gibco 15140-122) at 120,000 cells/cm^2^ cells on 35-mm or 60-mm culture dishes (Techno Plastic Products (TPP), Trasadingen, Switzerland). After 12h of culture, VCT were carefully washed to eliminate non-adherent cells. Purified VCT cultures were characterized by the ability to aggregate at 48 h of culture and to form ST at 72 h, and by following the production of hCG into the supernatant. Cells were harvested and frozen in liquid nitrogen and stored at −80 °C until RNA or protein extraction.

The choriocarcinoma human cell line BeWo derived clone, b30 (by Alan L. Schwartz) [67], was cultured in F-12K Medium (Kaighn’s Modification of Ham’s F-12 Medium, Gibco), supplemented with 10% FCS (Eurobio #CVFSVF00-01, Les Ulis, France), 2 mM l-glutamine, 50 IU/mL penicillin, and 50 µg/mL streptomycin. Cells were seeded at 12,000 cells/cm^2^ in 35-mm or 60-mm TPP culture dishes. Cells were collected at 24 h, 48 h, and 72 h for total RNA or protein extraction or fixed in 4% paraformaldehyde for further immunolabelling.

### 4.4. Other Cell Types

Mesenchymal cells (MC) from the first trimester villi were purified as previously described [68]. Artery (A) and veins (V) endothelial cells were obtained from first trimester umbilical cord after dissection. Heterogeneous cell mixture (HC) from first trimester umbilical cord obtained by type I collagenase composed of choriotrophoblasts and mesenchymal cells. Endothelial progenitor cells (EPC) purified from term placenta as previously described were graciously given by C. Boisson-Vidal [69].

### 4.5. Cell and Explant Incubation

Benzo-(a)-pyrene (B(a)P) (Sigma-Aldrich) was dissolved in DMSO at 40 mM, stored at 4 °C and used at a final concentration of 1 µM in culture medium. The final concentration of DMSO in the culture media was 0.0025%. All controls and incubated cultures contained the same amount of DMSO. CH223191 (Calbiochem, Darmstadt, Germany) was used at 3 µM in culture medium (stock at 3 mM in DMSO) and was incubated with cells for 1 h prior to B(a)P incubation.

### 4.6. RNA Extraction, Reverse Transcription, and Quantitative Real-Time PCR

BeWo cells in 35 mm petri dishes and placenta fragments stored at −80°C were lysed in TRIzol^®^. Total RNAs were then extracted following the manufacturer’s instructions using the Direct-zol RNA Miniprep kit (Zymo research, #R2052, Saint-Quentin-en-Yvelines, France) and quantified using a Nanodrop spectrophotometer. Reverse transcription was performed using the high-capacity cDNA reverse transcription kit (Applied Biosystems, Villebon sur Yvette, France) for 1 µg RNA. Gene-specific primers used for the real-time PCR were designed using the OLIGO Explorer software (Molecular Biology Insights, Hawthorne, NY, USA) (Table 3). Quantitative real-time PCR was carried out in a 10 μL reaction volume containing 1 µg of cDNA, 45 nM of each primer, and Takyon™ MasterMix (Takyon™ Rox SYBR^®^ MasterMix dTTP Blue, Eurogentec, Angers, France) using an ABI Prism 7900 Sequence Detector system (Applied Biosystems). PCR cycles consisted of the following steps: Takyon activation (3 min, 95 °C), denaturation (10 s, 95 °C), and annealing and extension (1 min, 60 °C). The threshold cycle (Ct) was measured as the number of cycles at which the reporter fluorescent emission first exceeds the background. The relative amounts of mRNA were estimated using the ΔΔCt method and then expressed as fold change. Primers for *HPRT*, *RPL13*, and *ubiquitin C* were used for the normalization of the results obtained with the BeWo cell line; *HPRT*, *RPL13*, and *18S*, then *KRT7*, for the normalization of the results obtained with placental villi, and *HPRT*, *RPL0, SDHA*, and *18S* for the normalization of the results obtained with trophoblasts. The results are given as fold change and each gene was normalized to the geometric mean of at least 3 reference genes that were found to be unchanged among the different conditions used.

### 4.7. RNA Extraction, Reverse Transcription and PCR

CT seeded for 24 h in 35 mm petri dishes were lysed in TRIzol^®^. Total RNAs were then extracted following the manufacturer’s instructions using the Direct-zol RNA Miniprep kit (Zymo research, #R2052) and quantified using a Nanodrop spectrophotometer. Reverse transcription was performed using the SuperScript III First Strand Synthesis system (Invitrogen 18080044, Villebon-sur-Yvette, France) for 500 ng RNA. PCR was carried out in a 50 μL reaction volume following the manufacturer’s instruction (GoTaq DNA polymerase kit, Promega M3005, Charbonnières les Bains, France) using an ABI Prism 7900 Sequence Detector system (Applied Biosystems). PCR cycles consisted of the following steps: Initial denaturation (5 min, 94 °C), denaturation (30 s, 94 °C), annealing (30 s, 57 °C), and extension (30 s, 72 °C). Amplification products were analyzed directly on 2% agarose gel and evaluated under UV light (Minibis, DNR Bio-imaging systems, Dublin, Ireland).

### 4.8. Cell Fractionation

Nuclear and cytoplasmic fractions, from BeWo cells or from chorionic villi, were prepared by a conventional differential centrifugation procedure using the NE-PER™ Nuclear and Cytoplasmic Extraction kit (Thermofisher, #78833, Villebon-sur-Yvette, France). Cells were harvested with trypsin-EDTA, centrifuged at 500× *g* for 5 min; then, pellets were collected, washed with PBS buffer, centrifuged again, and dry-pellets were then allowed to swell for 15 min in an ice-cold CER I buffer containing protease inhibitor cocktail (Sigma-Aldrich). Freshly harvested chorionic villi were dissected into approximatively 20–100 mg of tissue fragments, washed with PBS, and centrifuged at 500× *g* for 5 min. Dry-pellets were collected and homogenized in ice-cold CER I buffer at a volume of 200 µL for 20 mg tissue, or for 20 µL of collected cells. Cell disruption was performed by vortexing for 20 s (10 times) on the highest setting and verified under the microscope by staining the nuclei of broken cells with Trypan Blue (*v*/*v*). Then homogenates were incubated on ice for 10 min, CER II buffer added (11 µL), vortexed for 5 s, incubated on ice for another 1 min, vortexed again for 5 s, and centrifuged at 16,000× *g* for 5 min. The supernatant containing the cytoplasmic extracts were immediately transferred to a clean pre-chilled tube and stored at −80°C. Pellets containing nuclear fractions were suspended in ice-cold NER buffer (100 µL), vortexed for 15 s, placed on ice, and underwent vortexing for 15 s every 10 min, for a total of 40 min. Then homogenates were centrifuged at 16,000× *g* for 10 min and the supernatants containing nuclear extracts were stored at −80 °C for further analysis. Fraction purity was controlled by immunoblotting with α-Tubulin and Lamin A/C for the cytosolic and nuclear fractions, respectively.

### 4.9. Western Blot

Total protein extracts from BeWo cells or placental tissues were obtained by harvesting cells in Laemmli buffer (0.06 M Tris-HCl, pH 6.8, 10% glycerol, 2% SDS, protease inhibitors (Calbiochem), phosphatase inhibitor (Calbiochem)), heated for 1 min at 95 °C, and then sonicated. Protein concentrations were determined using the Pierce™ BCA Protein Assay Kit (Thermofisher). Equal amounts of proteins (40 µg) were separated on 4–15% SDS-PAGE mini-PROTEAN^®^ TGX™ precast protein gel under reducing conditions (DTT) and transferred onto a nitrocellulose membrane (Trans-blot Turbo Transfer pack, 0.2 µm nitrocellulose, Bio-rad, Marnes la Coquette, France). Blots were incubated overnight with the primary antibody at 4 °C, and then for 2 h with the appropriate DyLight 680 or 800 Fluor-conjugated secondary antibody (Thermo Scientific). The primary antibodies used were: Mouse monoclonal anti-AhR at 4 µg/mL (#WH0000196-M2, Sigma-Aldrich), mouse monoclonal anti-ARNT (A-3, sc-17811, Santa Cruz, Heidelberg, France), mouse polyclonal anti-TDO2 (SAB1406519, Sigma-Aldrich), rabbit monoclonal anti-IDO1 (86630, Cell signaling technology, Saint-Quentin-en-Yvelines, France), mouse monoclonal anti-α-Tubulin (#MS-581-P0, ThermoFisher), rabbit polyclonal anti-Lamin A/C (H-110, #sc-20681, Santa Cruz), mouse monoclonal anti β-Actin (A5441, Sigma-Aldrich), mouse monoclonal anti-vinculin (V9131, Sigma-Aldrich), and mouse monoclonal anti-CD31 (M0823, Dako, Les Ulis, France). Anti-AhR antibody at 4 µg/mL was also preincubated overnight at 4 °C with the corresponding antigen at 40 µg/mL (hereafter called blocking peptide, APREST78064, Sigma-Aldrich) in order to confirm its specificity. Secondary antibodies were IRDye 800CW-conjugated anti-mouse (#926-32212, Li-COR, Bad Homburg, Germany) or DyLight 680-labeled anti-rabbit (#35568, Thermo Scientific), and blots were scanned with an Odyssey^®^ Imaging System (Li-COR). Quantitation was performed using Li-COR Odyssey software. In some blot images, unnecessary lanes were cut off and clearly demarcated using black lines.

### 4.10. Tissue Immunolabelling

Placental villi fixed in 4% paraformaldehyde were included in 5% agarose, sliced in 100 µm sections using a vibratome and permeabilized in 0.5% triton X-100 in PBS for 30 min. Sections were saturated with a solution of 3% bovine serum albumin (BSA) and triton 0.1% X-100 in PBS for 4 h and then incubated overnight at 4 °C under agitation either with the primary antibody, with non-specific rabbit IgG1, or with a mixture of the anti-AhR antibody at 1 µg/mL and the corresponding antigen at 10 µg/mL (APREST78064, Sigma-Aldrich) that correspond to the sequence between 721 and 820 amino acids (C-terminus) of human AhR. Primary antibodies used were: 1 μg/mL anti-AhR antibody #WH0000196-M2, #A9359, #HPA029722, and #HPA029723 (Sigma-Aldrich), #Ab2770 (Abcam, Cambridge, United Kingdom), and 0.75 µg/mL anti-Cytokeratin 7 (#M7018, Dako, Les Ulis, France). After washing, the sections were incubated with Alexa Fluor 555 donkey anti-mouse and Alexa Fluor 488 goat anti-rabbit antibodies in 1% PBS-BSA for 2 h at room temperature. Nuclei were stained with DAPI for 5 min and Vectashield was used as mounting media for confocal microscopy images (Leica TCS SP2 confocal microscope, Leica Microsystems, Nanterre, France).

### 4.11. Cell Immunolabelling

At 24 h after seeding in 30 mm culture dishes (TPP), BeWo cells were fixed in 4% paraformaldehyde for 20 min, permeabilized in 0.1% Triton X-100 for 10 min, saturated with a solution of 1% BSA and 0.1% Tween in PBS for 2 h, and then incubated overnight at 4 °C with anti-AhR antibody as previously (1 μg/mL) or a mixture of 1µg/mL anti-AhR antibody (#WH0000196-M2) and 10 µg/mL of the corresponding antigen, and with 1 µg/mL anti-E-cadherin antibody (#EP700Y, Abcam). The primary antibodies were diluted in PBS containing 1% BSA and 0.1% tween. After washing, the cells were incubated with Alexa Fluor 555 donkey anti-mouse and/or Alexa Fluor 488 goat anti-rabbit antibody for 2 h at room temperature. Nuclei were stained with DAPI for 5 min and Dako fluorescence mounting medium was used for microscopy images. Images were taken by confocal or epifluorescence microscopy.

### 4.12. Statistical Analysis

The experiments were reproduced at least three times. For the ontogeny and for trophoblast differentiation, a minimum of *n* = 8 independent placentae were used. Quantified data are expressed as the means ± standard deviation. The mean values were analysed by a one-way ANOVA followed by a Mann Whitney’s test (Prism) and the level of significance was fixed at *p* < 0.05.

## 5. Conclusions

In conclusion, we have described here a complete spatio-temporal cartography of the expression and distribution of the AhR transcription factor—a key factor with several functions (detoxification and physiological processes)—in several human placental models (Figure 8). Our data strongly suggest a constitutive nuclear localization of AhR and its intrinsic activation in human term placenta due to the presence of some endogenous ligands produced by placental metabolism. Interference with this physiological function of AhR may explain pregnancy complications observed, for instance, after exposure to pollutants that activate the AhR pathway.

## Figures and Tables

**Figure 1 ijms-19-03762-f001:**
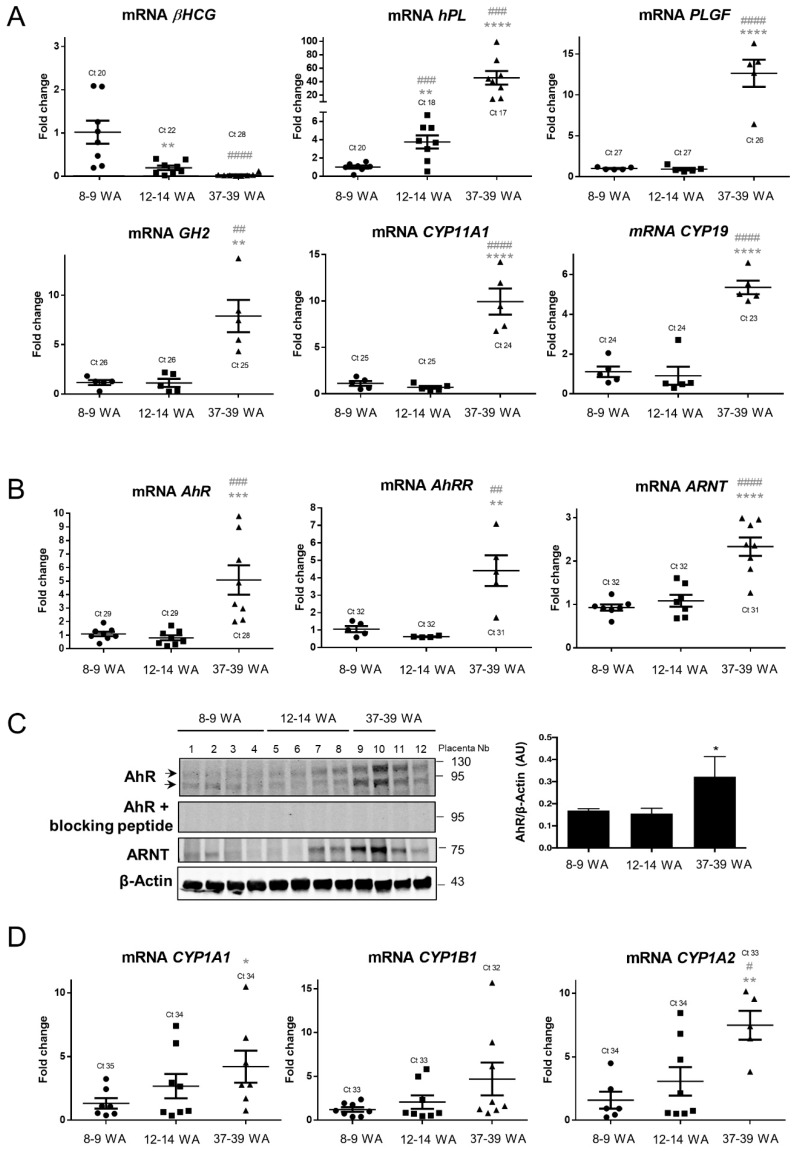
Placental expression of aryl hydrocarbon receptor (AhR) and relevant biomarkers during pregnancy. Total mRNAs were extracted from chorionic villi of eight placentae at 8–9 weeks of amenorrhea (WA), at 12–14 WA and at 37–39 WA (term). (**A**) Levels of *βhCG*, *hPL*, *PLGF*, *GH2, CYP11A1,* and *CYP19* were determined by RT-qPCR and normalized to the geometric mean of *HPRT*, *RPL13*, and *18S* transcripts, and reported to CK7 (*KRT7*). The means (long horizontal lines) and standard deviations (short horizontal lines) of eight independent experiments are shown. ** compared to 8–9 WA *p* < 0.01; ^##^ compared to 12–14 WA *p* < 0.01; ^###^ compared to 12–14 WA *p* < 0.001; **** compared to 8–9 WA *p* < 0.0001; ^####^ compared to 12–14 WA *p* < 0.0001. Ct is the raw value of the gene of interest without normalization to reference genes and to *KRT7*. (**B**) Levels of *AhR*, *AhRR*, and *ARNT* were determined by RT-qPCR and normalized to the geometric mean of *HPRT*, *RPL13*, and *18S* transcripts, and reported to CK7 (*KRT7*). The means (long horizontal lines) and standard deviations (short horizontal lines) of eight independent experiments are shown. ** compared to 8–9 WA *p* < 0.01; ^##^ compared to 12–14 WA *p* < 0.01; *** compared to 8–9 WA *p* < 0.001; ^###^ compared to 12–14 WA *p* < 0.001; **** compared to 8–9 WA *p* < 0.0001; ^####^ compared to 12–14 WA *p* < 0.0001. Ct is the raw value of interest gene without normalization to reference genes and to *KRT7*. (**C**) Total protein were analyzed by immunoblotting using AhR antibody alone or pre-incubated with its blocking peptide, AhR nuclear translocator (ARNT) antibody, and β-Actin, the last used as a loading control. The graph represents the total amount of AhR protein (2 bands) relative to β-Actin levels determined by quantification of immunoblot analysis using the Odyssey System Imager. The means (long horizontal lines) and standard deviations (short horizontal lines) of four distinct placentae per stage of pregnancy are shown (* *p* < 0.05). (**D**) Levels of *CYP1A1*, *CYP1B1*, and *CYP1A2* were determined by RT-qPCR as above. The means (long horizontal lines) and standard deviations (short horizontal lines) of eight independent experiments are shown. Ct is the raw value of interest gene without normalization to reference genes and to *KRT7*. * compared to 8–9 WA *p* < 0.05; ^#^ compared to 12–14 WA *p* < 0.05; ** compared to 8–9 WA *p* < 0.01.

**Figure 2 ijms-19-03762-f002:**
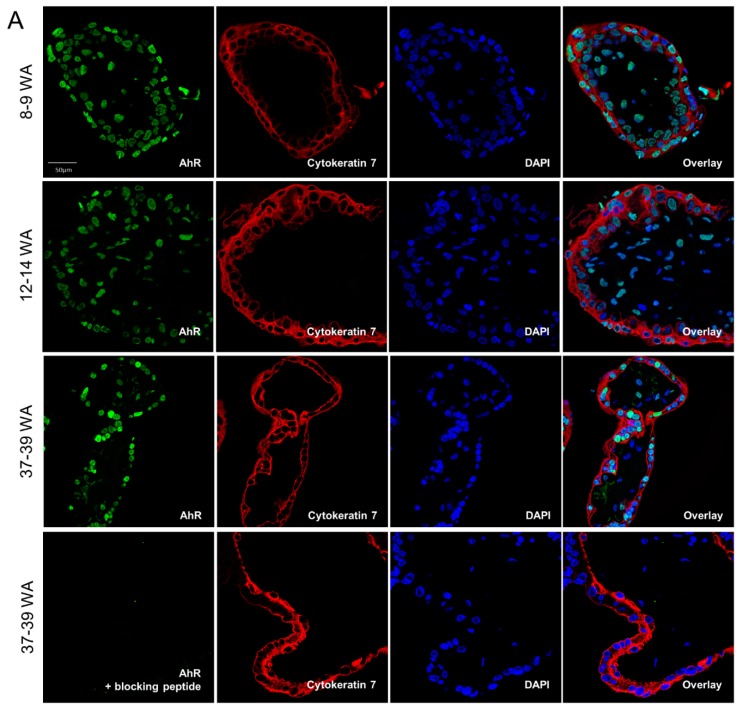
AhR localization in chorionic villi of placenta at different periods of pregnancy. (**A**) Chorionic villi from placentae for different periods of pregnancy (as indicated) were fixed in 4% PFA, included in agarose, and permeabilized with 0.5% Triton. Images are of confocal microscopy after immunostaining with anti-AhR antibody alone (HPA029723) (green, upper images) or anti-AhR pre-incubated with a blocking peptide (lowest image), of anti-cytokeratin 7 antibody (red, to underline the trophoblasts) and DAPI (blue, for nuclei). The overlay images are shown at right. Images are representative of four independent experiments. (**B**) Immunoblotting was carried out using anti-AhR antibody after cell fractionation of freshly harvested term chorionic villi to separate nuclear and cytosolic extracts. α-Tubulin and Lamin A/C are nuclear and cytosolic markers, respectively. N: Nucleus, C: Cytosol. The bar graph represents the total amount of AhR protein (two bands) normalized as following: Nuclear AhR level was normalized by Lamin A/C signal then nuclear contamination by cytosolic fraction (α-Tubulin/Lamin A/C ratio) was subtracted; cytosolic AhR level was normalized by α-Tubulin level then cytosolic contamination by nuclear fraction (Lamin A/C/ α-Tubulin ratio) was subtracted. Protein levels were determined by quantification of immunoblot analysis using the Odyssey System Imager. The means (long horizontal lines) and standard deviations (short horizontal lines) of three distinct villi are shown (*** *p* < 0.001).

**Figure 3 ijms-19-03762-f003:**
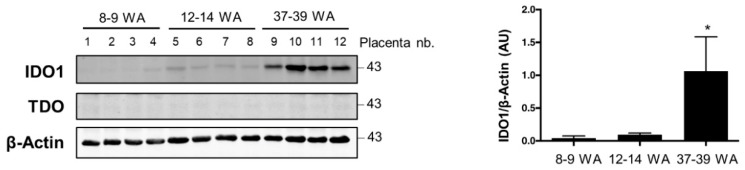
Indoleamine 2,3-dioxygenase 1 enzyme (IDO1) protein expression in placenta from different periods of pregnancy. Total protein extracts from placenta at different stages of pregnancy, as indicated, were analyzed by immunoblotting using anti-IDO1 and anti-TDO2 antibodies. β-Actin was used as a loading control. Quantification of IDO1 immunoblots relative to β-Actin was determined using the Odyssey System Imager and is shown in the bar scale graph. * compared to 8–9 WA, *p* < 0.05, *n* = 4 (lower panel).

**Figure 4 ijms-19-03762-f004:**
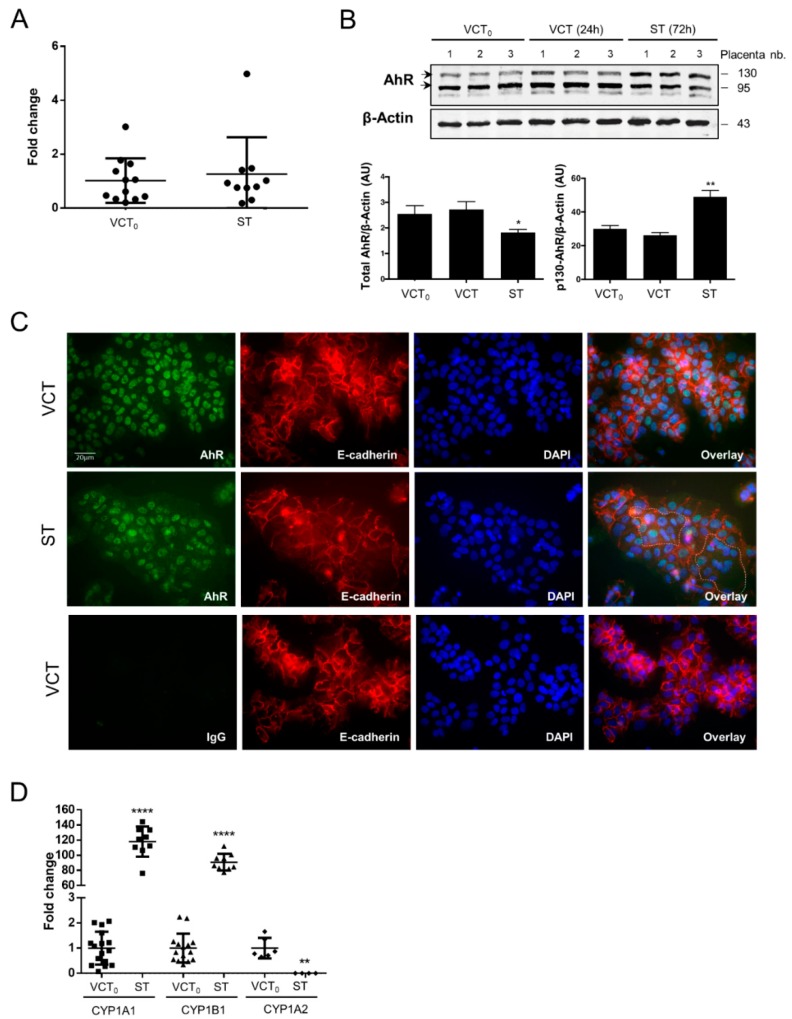
AhR expression and immunolocalization during CT differentiation into ST. (**A**) Total mRNA was extracted from freshly isolated trophoblasts before culture (VCT_0_) and after 72 h of culture when cells are differentiated into ST. The levels of *AhR* mRNA were determined by quantitative RT-qPCR. Data were normalized to *HPRT*, *RPL0*, *SDHA*, and *18S* mRNA levels. The means (long horizontal lines) and standard deviations (short horizontal lines) of at least *n* = 10 independent experiments are presented. (**B**) Western blot with protein extracts from purified trophoblast before culture (VCT_0_), after 24 h of culture (VCT), and after 72 h culture (ST). Membranes were immunoblotted with anti-AhR (WH0000196-M2 antibody) and β-Actin antibodies (loading control). Blot quantifications are shown in the bar graphs representing total AhR level (left panel) or upper 130-band (right panel) relative to β-Actin. * compared to VCT_0_, *p* < 0.05; ** compared to VCT_0_, *p* < 0.01. (**C**) Trophoblasts after 24 h of culture (VCT) and 72 h (ST) were fixed in 4% PFA and permeabilized with methanol. Images are of epifluorescence microscopy after immunostaining AhR (green), E-cadherin (membrane protein, in red), and nuclei (DAPI, blue). A control with non-specific IgG is shown in the lower panel and overlays in the right panel. (**D**) Freshly isolated cytotrophoblast before culture (VCT_0_) and after 72 h of culture (ST) were subjected to RT-qPCR after mRNA extraction. Levels of *CYP1A1*, *CYP1A2*, and *CYP1B1* are presented as the mean ± standard deviation of at least six independent experiments and normalized to *HPRT*, *RPL0*, *SDHA*, and *18S* mRNA levels. ** compared to VCT_0_, *p* < 0.05; **** compared to VCT_0_, *p* < 0.0001.

**Figure 5 ijms-19-03762-f005:**
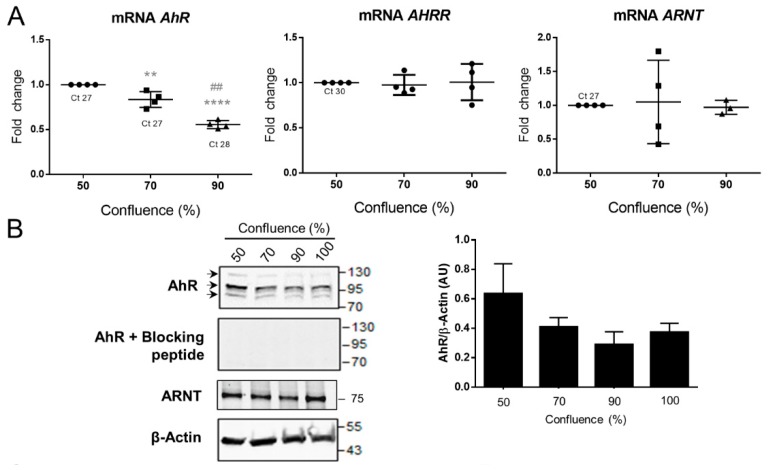
AhR expression, localization, and activity in the BeWo cell line. (**A**) Total mRNA were extracted from BeWo cells at various cell densities: 24 h hours after seeding cells reach about 50% of confluence; 48 h after seeding, they reached about 70% confluence, and about 90% 72 h after seeding. Levels of *AhR, AhRR,* and *ARNT* were determined by RT-qPCR and normalized to the geometric mean of *HPRT*, *RPL13*, and *Ubiquitin C* transcripts. The means (long horizontal lines) and standard deviations (short horizontal lines) of four independent experiments are shown. ** compared to 50% confluence *p* < 0.01; ^##^ compared to 70% *p* < 0.001; **** compared to 50% confluence *p* < 0.0001. Ct is the raw value of gene without normalization to reference genes (**B**) Total proteins were analyzed by immunoblotting using AhR antibody alone or pre-incubated with its blocking peptide, using ARNT and β-Actin antibodies, used as a loading control (left panel). The graph represents the amount of total AhR protein relative to β-Actin level determined by quantification of the immunoblot using the Odyssey System Imager (right panel). The means (long horizontal lines) and standard deviations (short horizontal lines) of three independent experiments are shown. (**C**) BeWo cells were fixed in 4% PFA and permeabilized with 0.1% Triton. Images are of epifluorescence microscopy after staining of AhR alone (green, upper images), AhR pre-incubated with a blocking peptide (middle images), or IgG (lower images) and DAPI (blue, for nuclei). The overlay images are shown at right. Images are relative to five independent experiments. (**D**) Immunoblotting was carried out using anti-AhR antibody after cell fractionation of BeWo cells to separate the nuclear and cytosolic extracts. α-Tubulin and Lamin A/C are nuclear and cytosolic markers, respectively. N: Nucleus, C: Cytosol. (**E**) Levels of *CYP1A1*, *CYP1B1*, and *CYP1A2* were determined and normalized to the geometric mean of *HPRT*, *RPL13*, and *Ubiquitin C* transcripts. The means (long horizontal lines) and standard deviations (short horizontal lines) of at least three independent experiments are shown. ** compared to 50% confluence *p* < 0.01; **** compared to 50% confluence *p* < 0.0001. Ct is the raw value of interest gene without normalization to reference genes.

**Figure 6 ijms-19-03762-f006:**
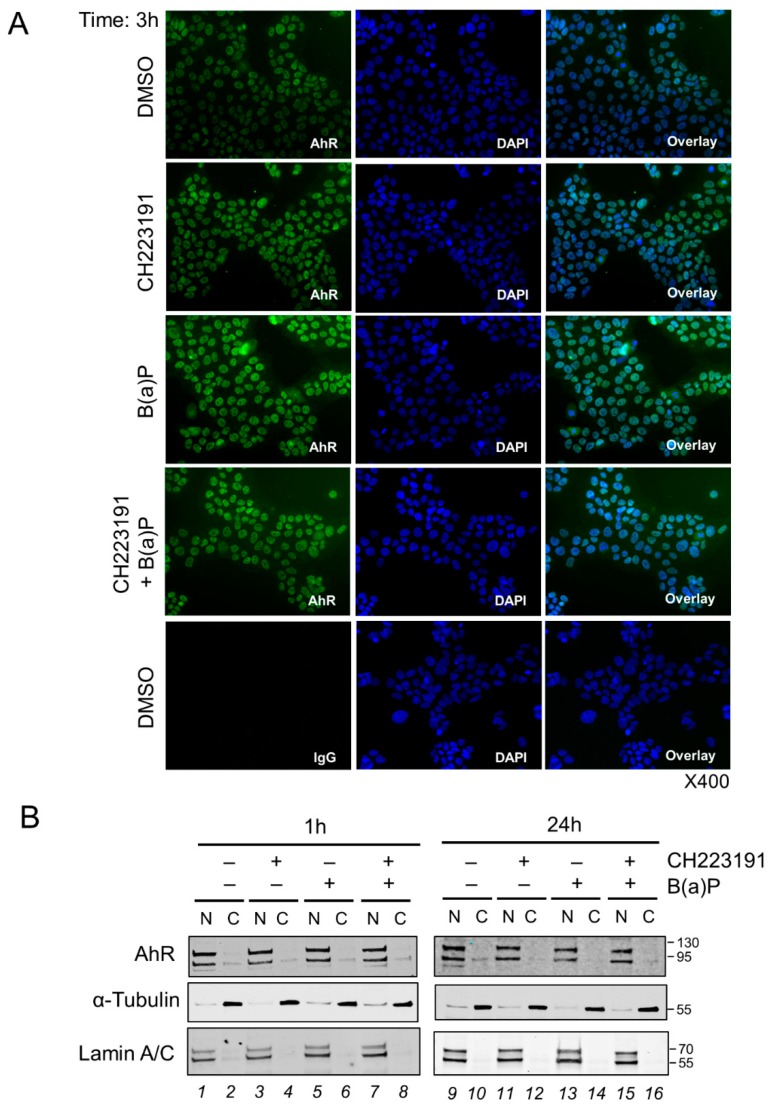
AhR localization in BeWo after incubation with benzo-(a)-pyrene. BeWo cells (at 70% confluence) were either used as control (solvent DMSO) or incubated with 1 µM B(a)P for the indicated times, in the presence or absence of a pre-incubation (1h before B(a)P) with 3 µM CH223191, an inhibitor of AhR, as indicated. (**A**) BeWo cells were fixed in 4% PFA and permeabilized with methanol. Images are of epifluorescence microscopy (400× magnification) after staining AhR with WH0000196-M2 antibody (green) and nuclei (DAPI). Overlays are in the right panel. (**B**) Cells were fractionated into nuclear (N) and cytosolic (C) fractions and subjected to Western blot. Immunoblotting was carried out using AhR (WH0000196-M2) antibody, and fractions purity was assessed with α-Tubulin (cytosolic) and Lamin A/C (nuclear).

**Figure 7 ijms-19-03762-f007:**
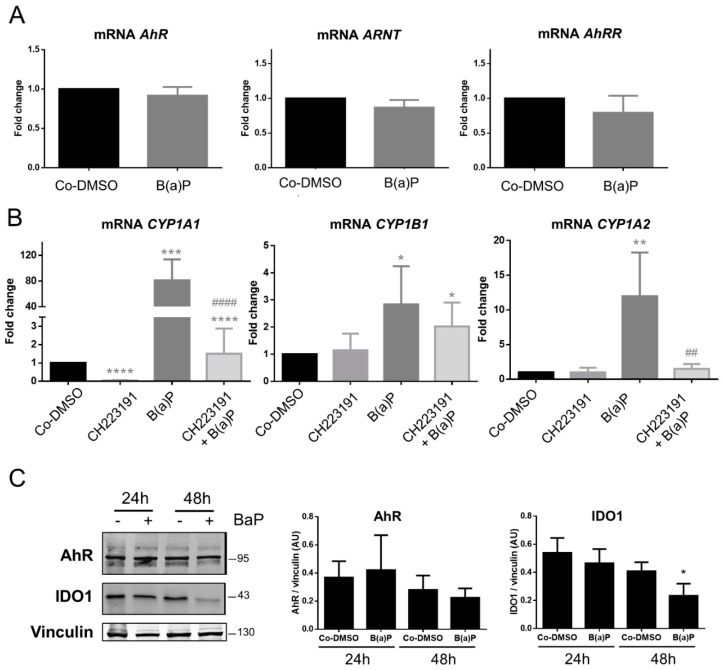
Impact of benzo-(a)-pyrene on AhR transcriptional activity. (**A**) Total RNA extracted from control BeWo cells (Co-DMSO) or incubated with 1 µM of B(a)P for 24 h, were analyzed by RT-qPCR using primers for *AhR*, *ARNT*, and *AhRR*. Results are normalized to three reference genes (*HPRT*, *RPL13*, and *Ubiquitin C*) and normalized to *CK7* (KRT7) and are presented as fold change. The means (long horizontal lines) and standard deviations (short horizontal lines) of three different experiments are shown. (**B**) BeWo cells (control (Co-DMSO) or cells incubated with 1 µM B(a)P for 24 h, in the presence or absence of 3 µM CH223191, a specific AhR inhibitor (1 h before B(a)P)), were subjected to RT-qPCR after total RNA extraction. Levels of *CYP1A1*, *CYP1B1*, and *CYP1A2* are presented as fold change, after normalization as above. * compared to Co-DMSO, ^#^ compared to B(a)P; **p* < 0.05; ** or ^##^
*p* < 0.01; *** *p* < 0.001; **** or ^####^
*p* < 0.0001 (*n* = 5). (**C**) Total protein extracts from chorionic villi (explants) of control term placenta (Co-DMSO) or incubated with 1 µM B(a)P for 24 and 48 h, respectively, were subjected to Western blotting. Immunoblotting was carried out with anti-AhR, anti-IDO1, and anti-Vinculin (as loading control). The bar graph represents the total amount of AhR protein (two bands) or IDO1 relative to Vinculin level determined by quantification of immunoblots using the Odyssey System Imager. The means (long horizontal lines) and standard deviations (short horizontal lines) of three different placentae at term are shown. * compared to DMSO, *p* < 0.05.

**Figure 8 ijms-19-03762-f008:**
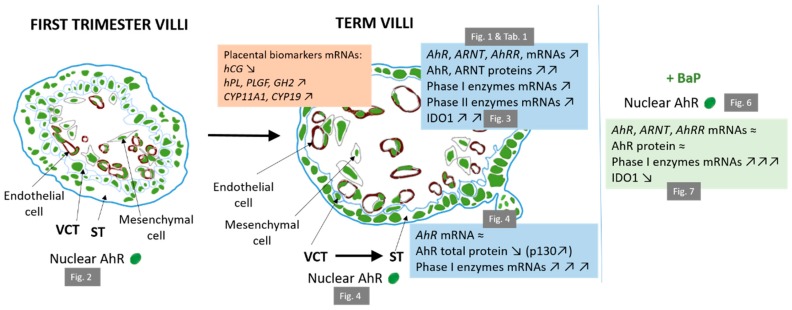
Summary of the results on AhR and relevant biomarkers’ changes during pregnancy. First trimester villi (on the left) evolve into term villi (on the right). Changes in placental biomarkers between the first trimester and term are found in the orange box, and changes in AhR, its partner ARNT, its repressor AhRR, phase I and II enzymes, and IDO1 between first trimester and term are found in the higher blue box. Changes in AhR and phase I enzyme during VCT differentiation into ST are found in the lower blue box. On the right side, changes in AhR, ARNT, AhRR, phase I enzymes, and IDO1 after incubation of BeWo cells or villi explants with 1 µM BaP are showed. Nuclear localization of AhR in the different cell types is visualized with a green bullet. Increase in a marker is represented by ↗, decrease by ↘, and no change by ≈.

**Table 1 ijms-19-03762-t001:** The expression of AhR-target genes involved in the oxidative defense system during ontogeny.

*mRNA*	Pregnancy Period	
	First Trimester (*n* = 16)	Term (*n* = 8)
	Fold Change Average ± Standard Deviation	
*HO1*	1.04 ± 0.33	2.24 ± 0.48 ***
*SOD1*	1.01 ± 0.16	3.99 ± 1.33 ***
*SOD2*	1.15 ± 0.64	3.09 ± 0.85 ****
*NQO1*	1.08 ± 0.51	3.53 ± 1.75 **
*GSTA2*	1.05 ± 0.39	2.42 ± 0.84 **
*Catalase*	1.06 ± 0.37	6.52 ± 2.62 ****

The means (long horizontal lines) and standard deviations (short horizontal lines) of 16 or eight independent experiments are shown. qRT-PCR data are normalized to the geometric mean of *HPRT*, *RPL13*, and *18S* transcripts, and reported to CK7 (*KRT7*) ** *p* < 0.01; *** *p* < 0.001; **** *p* < 0.0001.

**Table 2 ijms-19-03762-t002:** Expression of AhR-target genes involved in the oxidative defense system under varying BeWo cell densities.

*mRNA*	Confluence			
	70%		90%	
*HO1*	0.55 ± 0.06	****	0.37 ± 0.05	****, ##
*SOD1*	0.93 ± 0.05	*	0.64 ± 0.07	****, ###
*SOD2*	0.99 ± 0.13	-	0.56 ± 0.05	****, ###
*NQO1*	0.78 ± 0.11	**	0.40 ± 0.07	****, ##
*Catalase*	1.01 ± 0.12	-	0.66 ± 0.07	****, #
*GSTA2*	0.92 ± 0.14	****	0.66 ± 0.10	**, #
*GSS*	1.06 ± 0.14	-	0.77 ± 0.12	**, #
*NRF2*	1.02 ± 0.42	-	0.54 ± 0.15	***

The means and standard deviations (short horizontal lines) of eight independent experiments are shown. qRT-PCR data are normalized to the geometric mean of *HPRT*, *RPL13*, and *18S* transcripts, and reported to 50% confluence. * compared to 50% confluence *p* < 0.05; ^#^ compared to 70% confluence *p* < 0.05; ** compared to 50% confluence *p* < 0.01; ^##^ compared to 70% confluence *p* < 0.01; *** compared to 50% confluence *p* < 0.001; ^###^ compared to 70% confluence *p* < 0.001; **** compared to 50% confluence *p* < 0.0001, - no statistical difference.

**Table 3 ijms-19-03762-t003:** Primer sequences.

***AhR***	forward 5′-TAACCCAGACCAGATTCCTC-3′
	reverse 5′-GCAAACAAAGCCAACTGAG-3′
***AhRR***	forward 5′-CGCCTCAGTGTCAGTTACC-3′
	reverse 5′-ACTCACGACCAGAGCAAAG-3′
***ARNT***	forward 5′-CCCCTCCTGTAACCATTC-3′
	reverse 5′-CTGCCCACACCAAACTG-3′
**β*-actin***	forward 5′-CTCCTTAATGTCACGCACGATTTC-3′
	reverse 5′-ACAATGAGCTGCGTGTGGCT-3′
**β*hCG***	forward 5′-GCTACTGCCCCACCATGACC-3′
	reverse 5′-ATGGACTCGAAGCGCACATC-3′
***Catalase***	forward 5′-TTCATCCAGAAGAAAGCGGTCAA-3′
	reverse 5′-TGGATGTGGCTCCCGTAGTCA-3′
***CYP11A1***	forward 5′-TTTTTGCCCCTGTTGGATGCA-3′
	reverse 5′-CCCTGGCGCTCCCCAAAAAT-3′
***CYP19***	forward 5′-CCTGAAGCCATGCCTGCTGC-3′
	reverse 5′-CCGATCCCCATCCACAGGAATCT-3′
***CYP1A1***	forward 5′-CCACAGCACAACAAGAGAC-3′
	reverse 5′-CCATCAGGGGTGAGAAAC-3′
***CYP1A2***	forward 5′-TGCCAAACAGCATCATCTTG-3′
	reverse 5′-ACAGCACAACAAGGGACACA-3′
***CYP1B1***	forward 5′-AACGTACCGGCCACTATCAC-3′
	reverse 5′-CAGTGGTGGCATGAGGAATA-3′
***GH2***	forward 5′-CACCTTCCAACAGGGTGAAAA-3′
	reverse 5′-GGTGGCGATAGACGTTGCT-3′
***GSS***	forward 5′-GGAACATCCATGTGATCCGAC-3′
	reverse 5′-GGAACATCCATGTGATCCGAC-3′
***GSTA2***	forward 5′-TTGGGCTCTATGGGAAGGAC-3′
	reverse 5′-GGGAGATGTATTTGCAGCGGA-3′
***hPL***	forward 5′-GCATGACTCCCAGACCTCCTT-3′
	reverse 5′-TGCGGAGCAGCTCTAGATTGG-3′
***HO1***	forward 5′-CGTTCCTGGTCAACATCC-3′
	reverse 5′-CTGTCGCCACCAGAAAG-3′
***HPRT***	forward 5′-GGCGTCGTGATTAGTGATG-3′
	reverse 5′-CAGAGGGCTACAATGTGATG-3′
***KRT7***	forward 5′-GGACATCGAGATCGCCACCT-3′
	reverse 5′-ACCGCCACTGCTACTGCCA-3′
***NQO1***	forward 5′-GCAGACCTTGTGATATTCCAG-3′
	reverse 5′-CCTATGAACACTCGCTCAAAC-3′
***NRF2***	forward 5′-TGACATACTTTGGAGGCAAG-3′
	reverse 5′-CTAAATCAACAGGGGCTACC-3′
***PLGF***	forward 5′-GCTCGTCAGAGGTGGAAGTGGT-3′
	reverse 5′-CTCGCTGGGGTACTCGGACA-3
***RPL0***	forward 5′- AACATCTCCCCCTTCTCCT-3′
	reverse 5′- ACTCGTTTGTACCCGTTGAT-3′
***RPL13***	forward 5′- AAGGTCGTGCGTCTGAAG-3′
	reverse 5′-GAGTCCGTGGGTCTTGAG-3′
***SDHA***	forward 5′- CCACCACTGCATCAAATTCATG-3′
	reverse 5′-TGGGAACAAGAGGGCATCTG-3′
***SOD1***	forward 5′-CTGAAGGCCTGCATGGATTC-3′
	reverse 5′-CCAAGTCTCCAACATGCCTCTC-3′
***SOD2***	forward 5′-GTTCAATGGTGGTGGTCATA-3′
	reverse 5′-GTAAGTGTCCCCGTTCCTT-3′
***Ubiquitin* C**	forward 5′-CACTTGGTCCTGCGCTTGA-3′
	reverse 5′-TTTTTTGGGAATGCAACAACTTT-3′
***18S***	forward 5′- TCCCCCAACTTCTTAGAGG-3′
	reverse 5′-CTTATGACCCGCACTTACTG-3′

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
