# Peer review of "Expression, Localization, and Activity of the Aryl Hydrocarbon Receptor in the Human Placenta"

_ijms, 2018, doi:10.3390/ijms19123762_

Reviewer 1 Report

A very nice study.   A couple of comments for consideration but not necessary to address in order to finalize for publication:

a.  The paper could use a better explanation of how key placental marker genes ensure the viability of the placenta post harvesting.  I'm concerned that the harvesting conditions could result in tissue degradation that would impact the results.  Hence, having a good foundation of how viable placental tissue looks after harvesting (e.g., the expression of HCB, hPL, PLGF, GH2, etc.) will increase confidence that the AHR-related findings are related to non-degraded placental tissue.   In other words, do the placental biomarkers shown in Figure 1A match what others have reported for fold change in the literature.

b.  A diagram/cartoon of the placental anatomy/micro-anatomy where specific AHR-related changes were observed over the course of pregnancy, could be useful (in the discussion section)

c.  It would be interesting, using CHIP approaches, to examine the presence of ARNT, and/or other co-regulatory proteins, that are bound to the AHR receptor that is so highly localized in the nucleus.  Something for future studies - is the nuclear localized AHR sitting on a number of DREs and what does the complex look like?   With the AHR found to be highly localized in the nucleus, how does one establish whether or not it is simply sitting in the nucleus (like ARNT does) or if it is bound to specific DREs?

d.  Line 231 presents the kynurenine information - a few sentences describing how tryptophan is converted to kynurenine would be informative (for those who don't know this).  It is explained in the Discussion section but this information in the results seems to come out of no where for those unfamiliar with kynurenine and its linkage to the AHR and tryptophan.

e.   With respect to the nuclear localization, is it possible that this is an artifact induced by some unknown AHR that is present in one of the media (e.g., fetal calf serum or one of the other treatments) used to harvest and isolate sections of the placenta for AHR localization experiments?

f.  Could it be the 3 uM CH223191 concentration is simply not sufficient to overcome the endogenous ligand(s) that have much stronger binding affinities to the AHR.   Could a more potent AHR ligand inhibitor have been used?

g.  BaP did not change AHR localization (since most of the AHR is already in the nucleus) so what is the mechanism for BaP inducing CYP1A1/1A2 and CYP1B1.  One would assume that with maximal nuclear localization that CYP induction would be maximum in the placenta and yet BaP was able to induce mRNA?

Author Response

RESPONSES TO REVIEWER #1

Point a. The paper could use a better explanation of how key placental marker genes ensure the viability of the placenta post harvesting.  I'm concerned that the harvesting conditions could result in tissue degradation that would impact the results.  Hence, having a good foundation of how viable placental tissue looks after harvesting (e.g., the expression of HCB, hPL, PLGF, GH2, etc.) will increase confidence that the AHR-related findings are related to non-degraded placental tissue.  In other words, do the placental biomarkers shown in Figure 1A match what others have reported for fold change in the literature.

Answer to point a: As the Reviewer#1 pointed out, we now more clearly detail this part in the materials’ section for villi sampling of Figure 1. “The time for placental dissection was kept under 30 minutes, and tissues were snap frozen with liquid nitrogen, in order to limit tissue degradation [31].” (page 19, lines 630 and 632).

To clarify we modified the sentence in the results section with: “In order to verify that there are no ex vivo modifications, such as tissue degradation, during dissection or RNA extraction that could modify placental genes expression, we first assessed the level of typical placental biomarkers, such as β-hCG, hPL, PLGF, GH2, and enzymes, such as CYP11A1 (P450ssc) and CYP19 (Aromatase), responsible for the production of placental-derived progesterone and estrogens respectively [31] (page 3, lines 115-119).

Point b. A diagram/cartoon of the placental anatomy/micro-anatomy where specific AHR-related changes were observed over the course of pregnancy, could be useful (in the discussion section)

Answer to point b: We agree with the Reviewer #1’s comment for adding a cartoon that summarizes the results of the paper. We thus added the Figure 8 and Legend 8 (page 16, lines 447-456), and cited this new figure in the discussion (page 15, lines 437-438, 488, 568, 602 and 613).

Point c. It would be interesting, using CHIP approaches, to examine the presence of ARNT, and/or other co-regulatory proteins, that are bound to the AHR receptor that is so highly localized in the nucleus. Something for future studies - is the nuclear localized AHR sitting on a number of DREs and what does the complex look like?  With the AHR found to be highly localized in the nucleus, how does one establish whether or not it is simply sitting in the nucleus (like ARNT does) or if it is bound to specific DREs?

Answer to point c: We totally agree with Reviewer #1’s concerns about the study of AhR co-regulatory proteins that binds AhR when localized in the nucleus. We therefore added the following sentence in the discussion section: “In order to better understand the activity of AhR in the nucleus, it would be of high interest to further investigate the formation of complexes with its known partners, such as ARNT or other nuclear proteins, by performing chromatin immunoprecipitation in placental tissues and trophoblasts, in the absence of any voluntarily-added exogenous ligand” (page 17, lines 534-537).

Point d. Line 231 presents the kynurenine information - a few sentences describing how tryptophan is converted to kynurenine would be informative (for those who don't know this). It is explained in the Discussion section but this information in the results seems to come out of no where for those unfamiliar with kynurenine and its linkage to the AHR and tryptophan.

Answer to point d: We took into consideration Reviewer#1’s comment and completed the information with the following sentence in the introduction section: “Moreover, endogenous ligands like kynurenine (a L-tryptophan-derivate metabolite produced by two enzymes: the indoleamine 2,3-dioxygenase 1 enzyme (IDO1) or the tryptophan 2,3-dioxygenase (TDO)), which directly activate the AhR in immune cells from the decidua and peripheral blood, have been proposed to be essential for the maintenance of pregnancy, by allowing the immunological tolerance of the mother toward the embryo” (page 2, lines 87-89).

e. With respect to the nuclear localization, is it possible that this is an artifact induced by some unknown AHR that is present in one of the media (e.g., fetal calf serum or one of the other treatments) used to harvest and isolate sections of the placenta for AHR localization experiments?

Answer to point e: We agree with Reviewer #1 about the fact that AhR nuclear localization could be also due to the presence of serum from culture media. However, as we already mentioned in the discussion section (page 17, lines 524-531) the nuclear localization was backup by different techniques like tissue fractionation to separate nuclei from cytosols and was detected in freshly harvested tissue by immunolabeling with several anti-AhR antibodies. For these techniques, the chorionic villi were washed in HBSS and were not exposed to culture medium or fetal calf serum during the following steps of the protocol (fixation, permeabilization and staining). HBSS only contains ions and glucose, and these molecules are not known to be AhR activators. Consequently, it is unlikely that AhR nuclear localization in villi’s cells is an artefact. Concerning purified trophoblasts, indeed it would be interesting to investigate AhR localization in trophoblast cultures without fetal calf serum in the medium; hypothesising that trophoblasts are not subjected to serum-deprivation induced apoptosis (which should also be tested).

Point f. Could it be the 3 µM CH223191 concentration is simply not sufficient to overcome the endogenous ligand(s) that have much stronger binding affinities to the AHR. Could a more potent AHR ligand inhibitor have been used?

Answer to point f: CH223191 is a potent and selective AhR inhibitor commonly used in the literature at the micromolar range (Zhao et al., Toxicol Sci, 2010; Ghotbaddini et al., PloS One, 2017; Stanford et al., BMC Biology, 206). In our study, when BeWo cells were incubated with BaP, a strong AhR activator, CYPs mRNA expression was highly increased and the preincubation of cells with CH223191 partially or totally inhibited this induction. Consequently, this AhR inhibitor used at 3µM seems to be sufficient to overcome ligands that have strong binding affinity to AhR (Figure 7). However, as recommended by Reviewer #1, it could be interesting to investigate AhR activation using other AhR inhibitors such as: alpha-naphtoflavone, 1,3-dichloro-5-[(1E)-2-(4-chlorophenyl)ethenyl]-benzene, or 6,2',4'-Trimethoxyflavone. We now mention this AhR inhibitors in the discussion section: “In a next work it will be interesting to check other AhR-inhibitors such as alpha-naphtoflavone, 1,3-dichloro-5-[(1E)-2-(4-chlorophenyl)ethenyl]-benzene, or 6,2',4'-Trimethoxyflavone, because they have various selectivity of antagonisms” (page 18, lines 594-597).

Point g. BaP did not change AHR localization (since most of the AHR is already in the nucleus) so what is the mechanism for BaP inducing CYP1A1/1A2 and CYP1B1.  One would assume that with maximal nuclear localization that CYP induction would be maximum in the placenta and yet BaP was able to induce mRNA?

Answer to point g: According to reviewer #1’s concerns, we propose two hypotheses for BaP induced AhR-activation when AhR is already localized in the nucleus. As we mentioned in Fig.7 after BaP incubation of chorionic villi explants, we showed a drastic decrease in IDO1 protein level. In Fig.1 we showed a correlation of AhR expression and activity and IDO1 abundance; we therefore suggest that placental IDO1 levels could contribute to endogenous AhR activation by producing the kynurenine, an endogenous ligand of AhR. This is also supported by the increase in kynurenine levels observed in term placenta (Walker et al, Ref 49), as we mentioned in the discussion section (page 17, lines 494-496). Thus, BaP could lead to decreased production of an endogenous AhR ligand by decreasing IDO1 level. A second hypothesis would be competition between the endogenous (kynurenine) with the exogenous (BaP) ligands the last being in abundance at 1µM.

Reviewer 2 Report

This is an interesting paper on the important topic of localization and activity of AhR in human placenta.  The results are well presented, and have important implications regarding the role of this receptor in pregnancy.

There are some minor issues that should be addressed.  Some minor mistakes in the usage of English should be corrected.

The title for figure 1 should be changed to: "Placental expression of AhR and relevant biomarkers during pregnancy".  Also, in the legend to this figure, the authors use the abbreviation vs., e.g.,   ** vs 8-9 WA p < 0.01.  It is not clear what it means.

The title for figure 2 should be changed as well, to " AhR localization in chorionic villi of placenta at different periods of pregnancy."

Author Response

RESPONSES TO REVIEWER #2

Point 1: The title for figure 1 should be changed to: "Placental expression of AhR and relevant biomarkers during pregnancy".  Also, in the legend to this figure, the authors use the abbreviation vs., e.g.,   ** vs 8-9 WA p < 0.01.  It is not clear what it means.

Answer to point 1: As recommended by Reviewer #2, we change the title for Figure 1 (page 5, line 149) and clarify all the legends: “vs” was replaced by “compared to” in the whole article.

Point 2: The title for figure 2 should be changed as well, to "AhR localization in chorionic villi of placenta at different periods of pregnancy."

Answer to point 2: We completely agree with Reviewer #2 and changed the title for Figure 2 (page 7, line 214) as recommended.

Point 3: Some minor mistakes in the usage of English should be corrected.

Answer to point 3: As recommended by Reviewer #2, a native English speaker corrected English mistakes.